# ICD-11 Adjustment Disorder among Organ Transplant Patients and Their Relatives

**DOI:** 10.3390/ijerph16173030

**Published:** 2019-08-21

**Authors:** Rahel Bachem, Jan Baumann, Volker Köllner

**Affiliations:** 1I-Core Research Center for Mass Trauma, Tel Aviv University, Chaim Levanon 30, Tel Aviv 6997801, Israel; 2Bob Shapell School of Social Work, Tel-Aviv University, Chaim Levanon 30, Tel Aviv 6997801, Israel; 3Saarland University Medical Center, Faculty of Medicine, University of Saarland, 66421 Homburg/Saar, Germany; 4Department of Psychosomatic Medicine, Rehabilitation Center Seehof, Federal German Pension Agency, 14513 Teltow, Germany; 5Psychosomatic Rehabilitation Research Group, Department of Psychosomatic Medicine, Center for Internal Medicine and Dermatology Charité—Universitätsmedizin Berlin, 10098 Berlin, Germany

**Keywords:** adjustment disorder, ICD-11, organ transplantation, patients, relatives, somatic problems, quality of life, social support

## Abstract

Adjustment disorder (AD) is one of the most frequent mental health conditions after stressful life experiences in the medical setting. The diagnosis has been conceptually redefined in International Classification of Diseases (ICD-11) and now includes specific symptoms of preoccupations and failure to adapt. The current study assesses the prevalence of self-reported ICD-11 AD among organ transplantation patients and their relatives, explores the association of patients’ demographic-, transplant-, and health-related characteristics and ICD-11 AD symptoms, and evaluates the role of social support in the post- transplant context. A total of *N* = 140 patient-relative dyads were examined cross-sectionally. Hierarchical linear regression analyses were conducted to explore potential predictive factors of AD. The results revealed an AD prevalence of 10.7% among patients and 16.4% among relatives at an average of 13.5 years after the transplantation. The time that had passed since the transplantation was unrelated to AD symptom severity. Women tended to be at a higher risk in both groups. Somatic issues were predictive for AD only among patients and social support was predictive mainly among relatives. The results suggest that ICD-11 AD is a relevant diagnosis after organ transplantations for patients and relatives and its specific symptom clusters may provide important information for developing intervention strategies.

## 1. Introduction

### 1.1. Adjustment Disorder in ICD-11

An adjustment disorder (AD) is a maladaptive reaction to an identifiable and typically non- traumatic life event or stressful life circumstances, such as illness, disability, socioeconomic difficulties, or interpersonal conflict. AD was found to be one of the most frequently diagnosed mental health conditions in clinical practice [1], with particularly high prevalence rates in the liaison setting [2,3]. In the 11th revision of the International Classification of Diseases (ICD-11) [4], the diagnostic category of AD was included in a new chapter of stress-related disorders and has undergone major conceptual changes. For the first time, AD is defined as a full stress-response syndrome with specific diagnostic criteria rather than the narrative description of a sub-threshold disorder [5]. The two main symptom clusters of AD include preoccupations with the stressor (e.g., excessive worry, distressing thoughts and constant rumination) and failure to adapt (e.g., impaired daily functioning, sleep difficulties, loss of interest in work or social life, lack of the ability to recuperate) [6]. An AD is assumed to be a transient condition that resolves within six months after the stressor ended or within two years if the stressor persists for an extended period of time.

The recent ICD-11 revisions provide new options to describe and quantify stress-related symptoms that remain below the diagnostic threshold of established conditions such as post- traumatic stress disorder (PTSD), which may be particularly relevant in the medical context [7]. Patients with severe somatic illnesses have been reported to suffer from frequent subsyndromal PTSD [8,9], which is nevertheless associated with distinct impairments in life quality [10]. In such cases, the diagnosis of an AD may be applicable [11]. Moreover, as ICD-11 AD had been developed among a sample of patients with heart disease [12], it may be particularly well-suited to represent symptoms of AD after medical procedures. The current study thus explores the symptoms and correlates of ICD-11 AD in a sample of organ transplant patients and their relatives.

### 1.2. Organ Transplantation, Related Stressors and Long-Term Mental Health

Advancements in immunosuppressive treatment and surgical innovations have greatly improved the success rate and prognosis of organ transplant recipients’ physical health [13]. However, in the context of a transplantation, patients are exposed to a variety of severe stressors that include symptoms related to organ failure prior to the transplantation (e.g., life-threatening shortage of breath) and the intense fear of dying and extreme helplessness during the waiting period [14]. Moreover, the medical procedure itself often is experienced as a highly stressful or traumatic event [15,16]. After the transplantation, patients are faced with a new set of stressors such as the constant risk of transplant rejection, dangerous infections due to immune suppression medication, and other medical complications [8,13]. In the long term, the immune suppressive medication also increases the risk of cancer [17]. Therefore, the aftermath of an organ transplantation should be considered a permanent stress situation with the potential of debilitating mental health effects [9]. Organ transplantation can be seen as a paradigm for numerous other stressful experiences in the context of high-tech medicine (e.g., stem cell transplantation, implantation of a left ventricular assist device or an implantable cardioverter defibrillator).

Previous research on stress-response syndromes after organ transplantations has focused primarily on PTSD. A systematic review of the literature [9], reported prevalence rates of PTSD across different organ types from 1% at 3 months post-transplant [18] to 16% at a mean of 2.7 years post-transplant [19]. The prevalence in questionnaire-based studies tended to be more varied, ranging from 0% at 8 months post-transplant [20] to 46% at a mean of 1.1 years post-transplant [21]. In addition, a substantial number of subthreshold PTSD symptoms were reported, which ranged from 5% at 5 years post-transplant [13] to 20% at a mean of 3 years post-transplant [22]. Questionnaire studies seem to overestimate PTSD incidence which may be related to the fact that they partly include symptoms that would be more indicative of an AD [23]. Few studies, however, have assessed AD as a potential outcome of organ transplantations. In their pioneering study with heart transplant patients, Dew et al. (2001) conducted clinical interviews and established relatively high cumulative prevalence of AD according to the Diagnostic and Statistical Manual of Mental Disorders III after two months (6.6%), after one year (13.0%) and after three years (20.8%) [16].

The success of organ transplantations is not only evaluated by its effect on physical and mental health but also the patient’s post-transplant health-related quality of life [24]. Previous research has shown that life quality after organ transplantations tends to increase significantly compared to the pre-transplant state [25]. However, the presence of full PTSD as well as subsyndromal PTSD are related to lower health-related quality of life [13,24,26] in this subgroup of patients. Moreover, higher psychiatric problems in general, and PTSD in specific, were shown to be related to decreased treatment success, less adherence to medical treatment and higher mortality [9,22]. Given these significant adverse effect of stress-related mental health conditions, it is essential to better understand the factors that may be associated with AD in the post-transplant context. Thus, in the current investigation, the association of patient’s somatic problems and health-related life quality after transplantation with AD are being evaluated.

### 1.3. Relatives of Transplant Patients

Importantly, transplant patients do not live in a vacuum. The psychological distress related to a transplantation is likely to affect also their significant others [27]. Family members and friends are faced with the possibility of losing a loved one, which may be a potential trigger of AD [4]. Moreover, in the Diagnostic and Statistical Manual of Mental Disorders 5 [28] it has been acknowledged that indirect exposure to potentially traumatic situations, such as witnessing trauma or learning about a trauma that occurred to a loved one, can also cause PTSD. In an early work, Stukas et al. (1999) demonstrated that family caregivers indeed experienced PTSD at rates that were similar to PTSD among transplant recipients (10.5% among patients, 7.7% among caregivers) [29]. In one of the few studies that assessed AD among family caregivers, it was found that over the course of three years 34.5% of the relatives suffered from AD in at least one out of four assessment points [27]. These cumulative disorder rates among caregivers were even higher than the rates among the transplant recipient themselves [16,29]. The high psychological burden reported by significant others of organ transplant recipients is in line with a large literature on the aversive mental health effects of caregiving to family members with other chronic somatic illnesses [30,31]. However, there is currently little research available on AD among significant others of organ transplant patients and none that assess AD according to ICD-11.

### 1.4. Social Support in the Post-Transplant Context

Social and interpersonal factors are known to play a crucial role in recovering from traumatic and stressful life events [32] and perceived social support was shown to be one of the most important resources related to PTSD symptom severity [33,34]. Several studies have shown that also among transplant patients poor social support was associated with an increased risk of PTSD [22,35]. However, studies that assessed the role of social support for family caregivers reported mixed findings. Dew et al. (2004) [27] found that support from friends was unrelated to a higher risk of depression and anxiety disorders whereas Canning, Dew and Davidson (1996) [36] reported that more early post-transplant social support resources were associated with less distress both short-term and long-term. Moreover, recent research on AD outside the medical context has found that low perceived social support was associated with a higher probability of an AD diagnosis after involuntary job loss [37]. Given the potentially crucial role of social support during stressful life circumstances, the current study attempts to further clarify the role of social support among transplant patients and their relatives in association with ICD-11 AD.

To sum up, as the number of organ transplantations worldwide is increasing [38] and AD is a likely outcome among both patients and relatives, the current study explores AD symptomatology according to the new ICD-11 concept in a diverse sample of heart-, lung-, liver-, and kidney- transplant patients and relatives. The objectives of this study were: (1) to describe the prevalence of self-reported AD in a diverse sample of adult transplant recipients and their relatives and to compare AD and PTSD prevalence rates; (2) to explore the severity of AD symptoms among patients and relatives; (3) to assess the association of demographic-, transplant-, and health-related issues (i.e., somatic health, health-related quality of life) and transplant recipients’/their relatives’ AD symptoms; and (4) to assess the role of social support with regard to AD symptoms among transplant recipients and their relatives.

## 2. Materials and Methods

### 2.1. Participants and Procedure

The participants were members of a self-help association (Bundesverband der Organtransplantierten e.V.; BDO). The BDO is the largest self-help organization in Germany for individuals who are waiting for or have received an organ transplantation as well as for their relatives. It provides advice and care for patients and relatives before and after a transplantation, organises regional group meetings and doctor-patient seminars. Data were collected in June and July 2018. The study was approved by the ethics commission of the Saarland State Chamber of Physicians (registration nr. 108/18). The questionnaire in pen-and-paper format was sent to 600 members of the self-help association and their relatives. A total of 38.67% (*n* = 232) of transplant patients and 29.33% (*n* = 176) of relatives of transplant patients returned the questionnaire. The survey was additionally published online on the organization’s website and Facebook page. The online version was used by 18.10% (*n* = 42) of transplant patients and 11.93% (*n* = 21) of relatives. The current study relies on a subset of *n* = 140 patients and their relatives, for whom dyadic data was available. In cases where more than one relative of a transplant patient had participated in the study, the partner was chosen over other relatives as partners were assumed most likely to be primary caregivers and affected to a greater extent by their spouse’s illness than more distant family members. This led to the deletion of *n* = 3 parents, *n* = 4 children and *n* = 2 ‘others’.

### 2.2. Measures

#### 2.2.1. Adjustment Disorder

AD was assessed with the Adjustment Disorder—New Module (ADNM) [39]. The ADNM assesses symptoms of AD according to ICD-11 with 20 items and includes the two core groups of AD symptoms, “preoccupations” (4 items) and “failure to adapt” (4 items) as well as associated features of avoidance (4 items), depressive mood (3 items), anxiety (2 items), and impulse disturbance (3 items). Symptoms are rated on a 4-point Likert scale ranging from 1 (“never”) to 4 (“often”). Higher scores represent higher symptom severity. In the present study, we used a contextualized version and all items referred to the organ transplantation of the patient. Studies have confirmed convergent and discriminant validity of the ADNM-20 while test-retest reliability, internal consistency, and sensitivity to symptom change during treatment are good [40,41]. In the current study, internal consistencies were excellent (total score: α = 0.96, 0.97; preoccupations: α = 0.90, 0.90; failure to adapt: α = 0.85, 0.87, for relatives and patients respectively). For an ICD-11 diagnosis of AD, both preoccupations and failure to adapt symptoms causing significant functional impairment must be present. To approximate the ICD-11 diagnosis, the ADNM proposes a diagnostic algorithm [42], which requires that a participant scores ≥3 on at least one item and ≥2 on at least two items of both groups of core symptoms and rates impairment (“The symptoms cause clinically significant impairment in social, occupational, or other important areas of functioning”) ≥3.

#### 2.2.2. Posttraumatic Stress Disorder

PTSD was assessed according to ICD-11, using the PTSD- ICD- 11 questionnaire, which is a precursor of the International Trauma Questionnaire (ITQ) [43]. The ITQ was still under development when the current study commenced. The PTSD-ICD-11 had been used prior to the ITQ’s development, in international studies and showed good internal consistency (α = 0.76) [44,45]. It includes seven items relating to the experience of PTSD symptoms over the past month which are rated on a 4-point scale ranging from 1 (“never” or “up to once a month”) to 4 (“five times a week” or “almost always”). The main difference between PTSD-ICD-11 and ITQ is that the former assesses intrusive memories in two items (“Did you have disturbing memories about the event which came to mind uncontrollably even though you did not want to think about it?” “Was it as if you were experiencing the event again in the here and now?”) whereas the latter assesses it in one item (“Having powerful images or memories that sometimes come into your mind in which you feel the experience is happening again in the here and now?”). To approximate the format of the more recent ITQ, the two items related to disturbing memories in the PTSD-ICD-11 were summed up and averaged. In the present study, we used a contextualized version where all items referred to the organ transplantation of the patient. Internal consistency in the present study was satisfactory (α = 0.70).

In agreement with the ITQ diagnostic algorithm [43], a diagnosis of PTSD was assumed when one of two symptoms from the symptom clusters of (1) re-experiencing in the here and now, (2) avoidance, and (3) sense of current threat, plus endorsement of functional impairment were present. Endorsement of a symptom or functional impairment item is defined as a score >3 (“2–4 times per week or half of the time”).

#### 2.2.3. Somatic Problems

The severity of somatic symptoms was assessed with the Patient Health Questionnaire somatic symptom severity scale (PHQ-15) [46]. The PHQ-15 assesses the somatic burden with 15 items describing the most frequent consultation causes of physicians and frequent symptoms of somatisation disorder according to DSM IV. The items range from 0 (“not bothered at all”) to 2 (“bothered a lot”). Values of 5, 10 and 15 represent low, medium and high somatic burden. The PHQ-15 has been validated on primary care and obstetrics patients. Good internal consistency (Cronbach’s α = 0.80) was established in both samples and good convergent validity was reported with health-related life quality [46,47]. In the current sample, we found good internal consistency (Patients α = 0.83).

#### 2.2.4. Depression

Major depression was assessed with the Patient Health Questionnaire (PHQ-9) [48], which consists of nine items scored on a 4-point Likert scale ranging from 0 (“not at all”) to 3 (“nearly every day”) that correspond to the nine criteria of a DSM-IV diagnosis. Presence of at least 5 symptoms during more than half the days, one of which is either being in a depressed mood or the loss of interest, indicate probable depression. The PHQ-9 is a valid questionnaire with good psychometric properties [49].

#### 2.2.5. Health-Related Quality of Life

Health-related life quality was assessed with the Short Form Health Survey (SF-12) [50]. The SF-12 is a widely used self-rating instrument which measures patients’ health-related quality of life. The questionnaire consists of 12 items with heterogeneous rating scales ranging from a dichotomous response format (“yes” or “no”) to a 7-point Likert scale (“never” to “always”). The SF-12 provides two sum scores related to physical and mental health- related life quality. Higher values stand for better health-related life quality. Validation studies found good internal consistency for the physical (Cronbach’s α = 0.84) and mental (Cronbach’s α = 0.81) subscales [51], a clear 2-factor structure and good external validity [52].

#### 2.2.6. Social Support

Social support was assessed with the Oslo 3—Social Support Scale (OSLO) [53], an economic measure that is widely used in Europe and enquires after general social support in three items. Inquiries are made regarding the number of close friends, how involved other people were in the participants’ lives, and the availability of practical help from others (e.g., neighbours). The items add up to a sum score, which ranges between 3 and 14. The following categories can be distinguished: low (3 to 8), medium (9 to 11), high (12 to 14) [53]. The questionnaire has been used in several international studies that confirmed its predictive validity with respect to psychological distress [54,55]. In the current study internal consistency was satisfactory given the small number of items (α = 0.68 among patients, α = 0.67 among relatives).

### 2.3. Data Analysis

Data were analysed using IBM SPSS Statistics version 25 for Windows. Missing values occurred across variables and participants. Overall, 0.0%–12.1% of the data were missing. Little’s Missing Completely at Random (MCAR) revealed that the data were not missing completely at random, chi square (4904) = 5486.203, *p* = 0.00. Missing data were replaced with the Expectation Maximization algorithm, using all available data across scales for each participant to recover missing information [56].

A chi-square test was conducted to compare prevalence of AD between different groups. Gender differences in AD symptomatology and differences between patients and relatives were assessed via a *T*-test. Next, Pearson’s product-moment correlations were used to describe the relationships between AD of patients and relatives and all study variables (time since transplantation, age, PTSD symptoms, somatic problems of patients, physical health-related life quality of patients, and perceived social support of patients and relatives). Then, hierarchical linear regression analyses were conducted separately for patients and relatives. For patients, we inserted demographic information (gender, age) and transplant-related factors (time since transplantation, organ type) in a first step, health-related factors (somatic problems, health-related life quality) in a second step and social support in a third step. For relatives, we inserted demographics (age, gender) and time since transplantation in a first step, health-related factors of the patients in a second step and social support in a third step. We used an alpha level of *p* < 0.05 for determining the significance of statistical tests.

## 3. Results

### 3.1. Prevalence and Clinical Characteristics of AD

First, we evaluated the demographic and transplant-related characteristics of transplant patients and their relatives, which are shown in Table 1.

Second, we evaluated the prevalence of ICD-11 AD and PTSD separately for patients and relatives. Among patients, a total of *n* = 15 (10.7%) fulfilled the diagnostic criteria for AD whereas only *n* = 1 (0.7%) fulfilled the diagnostic criteria for PTSD. Women were twice as likely to suffer from AD (*n* = 10) than men (*n* = 5), though the difference was not statistically significant (*χ*^2^ (1) = 2.68, *p* = 0.087). If individuals who screened positive for major depression were excluded, a total of *n* = 12 (8.6%) of the patients screened positive for ICD-11 AD. Among relatives, a total of *n* = 23 (16.4%) fulfilled the criteria for AD whereas *n* = 3 (2.1%) reported symptoms corresponding to a diagnosis of PTSD. AD was more common among female relatives (*n* = 17) than male relatives (*n* = 6), though the difference was not statistically significant (*χ*^2^ (1) = 2.48, *p* = 0.180). If individuals who screened positive for major depression disorder were excluded, a total of *n* = 18 (12.9%) of the patients screened positive for ICD-11 AD. We then evaluated whether female and male participants differed in the severity of AD symptoms and found gender differences in the extent of a trend. Female patients (*M* = 33.92, *SD* = 14.703) tended to report more AD than male patients (*M* = 29.72, *SD* = 11.265) *t*(119.22) = 1.873, *p* = 0.064. Female relatives (*M* = 34.16, *SD* = 0.14.077), however, reported higher AD compared to male relatives (*M* = 29.31, *SD* = 11.668) *t*(120.62) = 1.71, *p* = 0.031. When combining patient and relative groups there was a significant gender difference. Women (*M* = 34.56, *SD* = 14.30) had higher AD than men (*M* = 29.55, *SD* = 11.39) *t*(276) = 2.925, *p* = 0.004.

Next, the interrelatedness of patients’ and relatives’ symptoms of AD were explored. A significant correlation of AD among both parties was found: *r* = 0.274, *p* = 0.001, which corresponds to a small effect size. Interestingly, a total of only *n* = 4 couples were identified where both patient and relative presented the full clinical picture of AD. Patients (*M* = 31.56, *SD* = 13.10) and relatives (M = 32.31, *SD* = 13.39) did not differ significantly in the severity of their AD symptoms *t*(278) = 1.096, *p* = 0.518. An analysis of group differences in the main symptom clusters of AD showed that relatives (*M* = 7.11, *SD* = 3.25) showed significantly higher preoccupations than patients (*M* = 6.23, *SD* = 2.84) *t*(266) = 6.42, *p* = 0.019. However, this difference was no longer significant if gender was a control variable *F*(1) = 0.250, *p* = 0.618. No significant difference was found for failure to adapt symptoms *t*(264) = 0.663, *p* = 0.241.

### 3.2. Correlations of Study Variables

Pearson correlation analyses are presented in Table 2 and revealed that patients’ higher AD was related to more somatic problems, more symptoms of PTSD, less physical health-related life quality and less perceived social support. Demographics (age and gender) or the time since transplantation were not correlated with patients’ AD. Relatives’ AD was related with their own PTSD symptoms and social support. Relatives’ AD was further associated with patients’ somatic problems, health- related quality of life and social support. The age of the relative but not of the patient was associated with the time since transplantation. Older patients reported lower health-related life quality. Not surprisingly, the patients’ somatic problems and health-related life quality were strongly interrelated. More somatic problems were associated with lower perceived social support of both patient and relative.

### 3.3. Predictors of AD in Patients and Relatives

Potential risk factors for AD were explored using stepwise regression analysis. The regression model for the patients was significant, *F*(7,117) = 14.314, *p* < 0.001, *R*² = 46%, and adjusted *R*² = 43% (see Table 3). Among the variables inserted in the first step, sex and organ type significantly contributed to the variance explained. Heart transplant patients had the highest AD symptoms (*M* = 34.12, *SD* = 14.858) and liver transplant patients had the lowest AD symptoms (*M* = 28.80, *SD* = 11.784). In the second step, somatic problems but not health-related life quality contributed significantly, beyond gender and organ type. Finally, social support contributed to the extent of a trend beyond all other variables but explained only an additional 1.4% of variance. No multicollinearity was found based on the variance inflation factor (VIF) (ranging between 1.053 and 1.835).

The regression model for the relatives was also significant, *F*(1,119) = 26.940, *p* < 0.001, *R*² = 27%, and adjusted *R*² = 23% (see Table 4). In the first step, only gender explained the variance in AD to the extent of a trend. In the second step, somatic problems and physical health-related life quality of the patients were added. Even though there was a significant increase in the variance explained (4.6%), the individual factors did not contribute significantly to relatives’ AD symptomatology. However, the effect of gender reached significance. In the third step, social support was added and contributed significantly to explaining an additional 17% of the variance in AD above and beyond the other factors. No multicollinearity was found based on the VIF measure (ranged between 1.061 and 1.682).

## 4. Discussion

The present study is the first to investigate AD according to the revised ICD-11 concept among a sample of post-transplant patients and their relatives. While a significant body of literature has dealt with the psychosocial consequences (e.g., PTSD) of organ transplantation for patients, relatively little is known about the psychological strains experienced by their significant others. The results suggest that ICD-11 AD seems to be a valid outcome of organ transplantations and its related adverse life circumstances, even years after the medical procedure. The disorder concerned relatives and patients to a similar extent, but women of both groups tended to be at a higher risk. Interestingly, the time that had passed since the transplantation was unrelated to AD symptom severity of both groups. Somatic problems were the most significant predictor of AD among patients but did not significantly predict AD among relatives. Although social support was a significant correlate of AD among patients, it explained little variance above and beyond the former risk factors. However, among relatives, social support explained a significant portion of variance beyond sociodemographic and patients’ health-related factors.

The results revealed that in the long-term aftermath of an organ transplantation, AD is a distinctly more prevalent clinical diagnosis than PTSD among both patients and relatives. This finding is in line with previous observations, stating that PTSD occurred almost exclusively early post-transplant and lasted for a medium duration of 7 months [29]. Given that the current study included a wide time frame after transplantation, the low prevalence of PTSD is not surprising. Moreover, the life-threatening traumatic experiences in the context of organ transplantations primarily occur during the waiting period before the medical procedure. It was established previously that in the long term, less dramatic health-related concerns and worries become central [14] and such difficulties are more likely triggers of AD.

Importantly, relatives were shown to be equally prone to AD symptoms as were the patients themselves. In the domain of preoccupations with the transplantation, a main symptom cluster of ICD-11 AD, relatives reported even stronger symptoms than the patients themselves. This finding is in line with a study that assessed PTSD in patients and their spouses after the implantation of a ventricular assist device [57]. The latter study identified PTSD exclusively among spouses, who worried more strongly about device malfunctions, infections, pain, or the danger of a stroke than did the patients themselves [57]. However, such cognitions correspond better with the ICD-11 AD symptom cluster of preoccupations, which delineates excessive worry, distressing thoughts and constant ruminations that centre around the critical life event or its consequences than with the clinical picture of PTSD. Future research that explicitly focuses on the distinctions of AD and PTSD symptoms after organ transplantations is indicated, particularly among relatives.

Interestingly, the correlation of AD symptoms between patient and relative was significant but small. This finding suggests that the organ transplantation and its sequalae are perceived and processed individually by patients and significant others. Empirical evidence from previous studies shows that patients and relatives experience different problems in the post-transplantation context. For example, a study on dyadic difficulties after heart transplantation found that whereas patients primarily experienced difficulties in talking about emotional matters (i.e., either failing to talk about problems at all or becoming overly emotional), partners emphasized more diverse difficulties in adapting to new roles in the family, communication difficulties with the afflicted family member, and a discrepancy between the patients’ interests and those of their spouses [58]. It may thus be suggested that different constellations of risk and protective factors among patients and relatives are more central to AD symptomatology in both groups than the characteristics of the shared adverse situation (i.e., transplantation aftermath).

Our analysis of demographic characteristics as potential risk factors yielded results only for female gender. Among patients, women were twice as likely to report the full clinical picture of AD than men, even though the difference in symptom severity remained a trend. This may have been due to the relatively small number of individuals in the current study that portrayed full-threshold AD. Nevertheless, the indication of a gender effect corresponds with research on other stress-related disorders (i.e., PTSD) in organ transplant patients [29] and with the literature on AD after other stress- events such as job loss [59,60]. Among relatives, however, gender differences were more pronounced and predicted a significant portion of the variance in AD symptomatology. Similar gender differences were found in previous studies among relatives of transplant patients [27,29]. It has been observed that in most societies, women tend to have fewer options for resisting the pressure to provide care than do men. Women may therefore be more likely to experience the ambivalence that results from pressure to provide care [61], which may ultimately manifest in higher symptoms of AD. The observation of small but significant gender differences in psychopathology further aligns with observations from a meta-analytic analysis on the effect of caregiving to chronically ill family members and found that women provided more caregiving hours and more personal care and reported higher levels of depression and lower levels of subjective well-being than men [62].

This study covered a wide range of time since the transplantation but no direct relationship with AD symptoms was established, neither for patients nor for relatives. Overall, this finding complements the literature on long-term development of PTSD symptoms and general distress after organ transplantations, which also were unrelated to the time since the medical procedure [9]. One previous study that pursued a similar research question with regard to AD symptoms according to ICD-11, investigated patients 0.3 to 10 years after the operation of a thoracic aneurysm and reported similar non-significant results for the association of time and AD [63]. However, as participants of the current study were recruited from among the members of a self-help organization, individuals who were suffering from long-term physical and/or mental health problems after the transplantation were likely overrepresented. Future studies should investigate trajectories of AD symptoms after transplantation in more representative samples.

However, the lack of association of AD symptom severity and time since transplantation is itself an important finding. AD is assumed to be a transient disorder that resolves spontaneously within six months after a stressor ended or within two years if a stressor persists, as is the case with chronic illnesses [4]. The current data suggest that this time limitation may not adequately describe the course of ICD-11 AD symptoms which for some individuals seem to persist even decades after the transplantation. The conceptualization of AD as a time-limited mental health diagnosis has been questioned elsewhere [60,64] and the current study adds empirical support to the debate around re- evaluating the AD time criterion.

Somatic symptom burden but not health-related life quality was the main predictor of AD symptoms among patients. This result illustrates that post-transplant complications such as transplant rejection or infections and other long-term health risks [8,13] are the most likely factor that sustains long-term mental health impediments. The literature documents that patients who do not have such complications after the transplantation return to pre-illness life quality [25] and also report lower symptoms of PTSD [14]. Moreover, the data revealed that the likely causal factor for AD symptoms among patients is the health issues themselves and not the associated impairments in health- related life quality. For the relatives, however, somatic problems and health-related life quality of the patient, even though correlated with AD, did not predict AD beyond demographic characteristics. Even though the organ transplantation seems to be the trigger of AD symptoms of the relative (relatives’ AD symptoms were assessed with reference to the transplantation), the related somatic complaints of the patient do not account for the relatives’ AD symptoms in the long run. This is consistent with the finding that patients’ and relatives’ AD symptoms were correlated to a small extent. Future research is needed to explore in detail the explanatory factors of AD among significant others.

Finally, we found that social support reported by patients explained only a small portion of variance beyond demographic and health factors. Given that social support is among the most established protective factors following stressful life events [34], this finding is surprising at first sight. However, the measure employed in this study assessed general social support by friends and acquaintances but did not explicitly include family members. As most of the care is provided by family members [27], it is possible that social support outside the family may be less crucial for patients’ transplant-related AD symptoms in the long term. Future studies should therefore assess whether intra-family social support and patient-caregiver relationships are more distinctly related to AD. Among patients, contrarily, general social support explained a significant portion of the variance in AD. This finding is not surprising as when a close family member is the recipient of care, the caregivers may particularly depend on other social resources such as family-external social support in order to cope with the illness of their significant other. Previous research shows, for example, that social support helps caregivers of chronically ill or older adults cope with caregiver burden [62,65].

The findings of this study should be considered in the light of several limitations. First, the sample was recruited via a self-help organization and therefore may tend to consist of patients who are particularly burdened by the experience, which may have resulted in an overestimation of the AD prevalence and limits the generalizability of the result. Moreover, the response rate was moderate, which resulted in a limited sample size. Second, AD was assessed using a self-report screening questionnaire rather than a clinical or a semi-structured interview. Third, from among the exclusion diagnoses mentioned in the ICD-11, we were only able to consider depression but did not have measures of prolonged grief disorder, uncomplicated bereavement, burn-out, and acute stress reaction, which may have resulted in an overestimation of AD prevalence rates. Fourth, the cross- sectional nature of the data does not allow for any inferences on causality. Fifth, the time since the transplantation varied widely. Even though this represents the reality of post-transplant patients, future research should determine the prevalence of ICD-11 AD in a more homogenous sample at a briefer interval after the procedure. Nevertheless, given the paucity of research on AD among transplant recipients in general and the ICD-11 concept in specific, this first examination yielded important exploratory information.

## 5. Conclusions

In conclusion, this study found that ICD-11 AD is a potential outcome of organ transplantations. The findings suggest that not only the patients themselves but also relatives are afflicted with psychological distress and should be considered as potential recipients of mental health services. Future research in the field of severe and chronic somatic disease should pay attention to identifying and evaluating strategies to support patients and relatives in coping with their preoccupations and failure to adapt symptoms. Such interventions may make use of personal coping resources, such as positive self- and world-views, religion, financial resources, and social resources, which have been shown to be associated with better psychological adjustment after organ transplantations and other stressful life events [66,67,68]. In the current study, the beneficial effect of family-external social support for relatives was particularly striking. Intervention groups held locally in transplantation centres or accessible online could address unmet needs for social support. Finally, it is noteworthy that the symptom burden of AD may persist in the long-term course after organ transplantation. This result implies that AD’s time criterion of two years should be evaluated in further studies of a longitudinal design.

## Figures and Tables

**Table 1 ijerph-16-03030-t001:** Demographic characteristics of organ transplant patients and their relatives.

Age (Years): M (SD) Range		Patients (*n* = 140)		Relatives (*n* = 140)	
		54.16 (16.07) 16–82		57.53 (11.49) 24–80	
		*n*	%	*n*	%
Age group	<50	48	34.3	29	20.7
	50 to 65	52	37.1	71	50.7
	>65	39	27.9	35	25.0
	missing	1	0.7	5	3.6
Sex	Women	65	46.4	88	62.9
	Men	74	52.9	51	36.4
	Missing	1	0.7	1	0.7
Employment	Employed	37	26.5	64	45.8
	Self-employed	9	6.4	12	8.6
	Homemaker	4	2.9	8	5.7
	Student	9	6.4		
	Retired	75	53.6	51	36.4
	Unemployed	2	1.4	3	2.1
	Other	3	2.1	2	1.4
	Missing	1	0.7		
Education	Junior High	24	17.1	31	22.1
	High School	70	50.0	72	51.4
	College	43	30.7	36	25.7
	Other	3	2.1	1	0.7
Organ transplant *	Liver	23	16.4		
	Lung	49	35.0		
	Kidney	26	18.6		
	Heart	49	35.0		
Time since transplantation (years)		13.5			
Relationship with patient	Partner			102	72.9
	Parent			25	17.9
	Child			9	6.4
	Other			3	2.1
	Missing			1	0.7

Note. ***** multiple selections possible.

**Table 2 ijerph-16-03030-t002:** Intercorrelations between main study variables.

Measure	1	2	3	4	5	6	7	8	9	10	11
*M* (*SD*)	31.60 (13.10)	32.31 (13.39)	7.12 (1.92)	7.45 (2.39)	161.29 (109.69)	54.1 (16.07)	57.5 (11.59)	7.77 (5.31)	46.21 (10.45)	9.56 (2.13)	9.31 (2.15)
1. Adjustment disorder, pat.	-										
2. Adjustment disorder, rel.	0.27 **	-									
3. PTSD, pat.	0.66 **	0.20	-								
4. PTSD, rel.	0.20 *	0.71 **	0.11	-							
5. Time since transplantation	0.03	0.09	−0.01	−0.01	-						
6. Age, patient	0.01	−0.02	−0.09	−0.03	0.14	-					
7. Age, relative	0.11	0.12	−0.07	0.04	0.24 **	0.51 ***	-				
8. Somatic problems, pat.	0.62 ***	0.17 *	0.43 **	0.18 *	0.14	0.15	0.06	-			
9. Physical life quality, pat.	−0.43 ***	−0.20 *	−0.25 **	−0.20 *	−0.08	−0.40 ***	−0.31	−0.58 ***	-		
10. Social support, pat.	−0.23 ***	−0.17 *	−0.26 **	−0.02	−0.21 *	−0.01	0.01	−0.32 ***	0.12	-	
11. Social support, rel.	−0.15	−0.39 **	−0.10	−0.24 **	−0.03	0.03	0.04	−0.18 *	0.04	0.39 ***	-

Note. *** *p* < 0.001 ** *p* < 0.01 * *p* < 0.05; pat. = patients; rel. = relatives; PTSD = post-traumatic stress disorder.

**Table 3 ijerph-16-03030-t003:** Hierarchical regression to predict AD among patients.

	b	se	β	t	*p*	*R*²ᶺ	*F*ᶺ	*P*ᶺ
Step 1								
Gender	−0.642 **	0.235	−0.253 **	−2.732	0.007			
Age	0.037	0.070	0.047	0.5326	0.600			
Time since transplantation	0.006	0.010	0.056	0.619	0.537			
Type of organ	2.174 *	1.025	0.191 *	2.121	0.036	0.00	2.640 *	0.037
Step 2								
Gender	−3.790 *	1.865	−0.149 *	−2.033	0.044			
Age	−0.073	0.061	−0.094	−1.200	0.233			
Time since transplantation	−0.005	0.008	−0.039	−0.546	0.586			
Type of organ	1.730 *	0.840	0.152 *	2.153	0.033			
Somatic problems	1.320 ***	0.202	0.562 ***	6.538	0.000			
Physical life quality	−0.134	0.111	−0.112	−1.211	0.228	0.366	39.050 ***	0.000
Step 3								
Gender	−3.338 ^	1.866	−0.131 ^	−1.789	0.076			
Age	−0.070	0.060	−0.091	−1.172	0.244			
Time since transplantation	−0.008	0.008	−0.067	−0.919	0.360			
Type of organ	1.718	0.796	0.151	2.157	0.033			
Somatic problems	1.216 ***	0.209	0.518 ***	5.829	0.000			
Physical life quality	−0.151	0.110	−0.126	−1.373	0.172			
Social support	−0.783 ^	0.443	−0.131 ^	−1.768	0.080	0.014	3.127 ^	0.080

Note. *F*ᶺ = F statistic of the change from previous step, *P*ᶺ = *p* value of the F change; ^ *p* ≤ 0.08 * *p* < 0.05, ** *p* < 0.01, *** *p* < 0.001.

**Table 4 ijerph-16-03030-t004:** Hierarchical regression to predict AD among relatives.

	b	se	β	t	*p*	*R*²ᶺ	*F*ᶺ	*P*ᶺ
Step 1								
Gender	−4.587 *^*	2.337	−0.176 *^*	−1.962	0.052			
Age	0.163	0.098	0.151	1.655	0.101			
Time since transplantation	0.001	0.011	0.006	0.063	0.950	0.056	2.406 ^	0.071
Step 2								
Gender	−5.367 *	2.320	−0.026 *	−2.314	0.022			
Age	0.114	0.103	0.105	1.105	0.271			
Time since transplantation	-0.002	0.011	−0.018	−0.199	0.843			
Somatic problems	0.293	0.253	0.125	1.158	0.249			
Physical life quality	−0.555	0.136	−0.128	−1.142	0.256	0.046	3.100 *	0.049
Step 3								
Gender	−6.853 **	2.123	−0.263 **	−3.288	0.002			
Age	0.129	0.093	0.120	1.383	0.169			
Time since transplantation	−0.003	0.010	−0.026	−0.304	0.761			
Somatic problems	0.068	0.233	0.037	0.370	0.712			
Physical life quality	−0.186	0.123	−0.153	−1.507	0.135			
Social support	−2.421 ***	0.466	−0.491 ***	−5.190	0.000	0.166	26.940 ***	0.000

Note. *F*ᶺ = F statistic of the change from previous step, *P*ᶺ = *p* value of the F change; ^ *p* ≤ 0.07 * *p* < 0.05, ** *p* < 0.01, *** *p* < 0.001.

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
