# Peer review of "ICD-11 Adjustment Disorder among Organ Transplant Patients and Their Relatives"

_ijerph, 2019, doi:10.3390/ijerph16173030_

Round 1

Reviewer 1 Report

Review of IJERPH- AD & transplantation

Many thanks for sending me this paper to review: adjustment disorder (AD) is a common and under-researched condition. In particular there is a need to consider in a transplant population where there is a significant stressors. This study is novel in considering a relative of the carer in parallel.

Main point:

This study finds that ~10% of the transplant population meet criteria on ADMN for AD, but this is a screening measure as opposed to a diagnostic measure (using either semi-structured interview or clinical diagnosis). Given that the mean time since transplant is 13.5 years when the cut-off for AD is 2years, this is unlikely to be AD, but instead some other phenomenon such as depressive episode or an anxiety disorder. The biggest limitation of this study is lack of any convincing diagnosis, and absence of any measure of depressive or anxiety symptoms, particularly when depression has been found to be present in post-transplant patients with a similar prevalence rate of up to 63% [1-4]. Given that one of the biggest controversies in AD research is the difficulty in distinguishing it from depression, this is certainly a point that is worthy of mention [5]. I remain unconvinced that AD was measured in this study.

General comments:

A.      Both the title and the abstract were, on the whole, well written and reflect the content of the paper proper.

B.      The introduction gives a good overview of the literature and where this study fits into this.

C.      The methods give a detailed overview of the study.

1.       Cross-sectional study

2.       Participants were recruited using self-reported questionnaires - a non-clinical population.

3.       The measures and the statistical analysis were appropriate and well described, although there were some critical gaps: the lack of diagnosis (clinical or semi-structured), and absence of any measure of depressive or anxiety symptoms

 4. The sociodemographic and clinical characteristics were described in the 'methods' section. Would these not be more appropriately detailed under 'results'?

D.      The results give a brief description of the findings.

E.       The discussion provides a reasonable summary of this study, and covers the area of comparison between this study and the existing literature well. The limitations are discussed in reasonable detail, and the authors consider the recruitment bias inherent to its methodology. However, the most obvious limitation remains unconsidered – the lack of diagnosis, and absence of any measure of depressive or anxiety symptoms.

F.       The conclusion is appropriate to the findings of the paper, although I do not believe that the finds of one cross-sectional study support the final sentence “Should these findings be replicated in a study of a longitudinal design, AD’s time criterion of two years should be reconsidered.” This sentence would be best omitted.

G.     The references are profuse, relevant and up to date.

H.      On the whole the tables and figures, are clear. In Table 1, mean and range for time since transplantation would have been a welcome addition.  Table 4 is titled “Hierarchical Regression to Predict AD among Patients”, when presumably it should read: “Hierarchical Regression to Predict AD among Relatives”.

I.        Minor typographic errors throughout, especially multiple counts of “infect” for “infection”.

References

1.  Dew MA, DiMartini AF. Psychological disorders and distress after adult cardiothoracic transplantation. J Cardiovasc Nurs. 2005; 20(5 Suppl): S51.

2.  Fusar-Poli P, Lazzaretti M, Ceruti M, et al. Depression after lung transplantation: causes and treatment. Lung. 2007; 185: 55.

3. Zalai D, Szeifert L, Novak M. Psychological distress and depression in patients with chronic kidney disease. Semin Dial. 2012; 25: 428.

4. Dew MA, Rosenberger EM, Myaskovsky L, et al. Depression and Anxiety as Risk Factors for Morbidity and Mortality After Organ Transplantation: A Systematic Review and Meta-Analysis. Transplantation. 2015; 100: 988–1003.

5. Bachem R, Casey P. Adjustment disorder: A diagnosis whose time has come. J Affect Disord, 2018; 227: 243-253.

Author Response

Response to Reviewer 1 Comments

Review of IJERPH- AD & transplantation

Many thanks for sending me this paper to review: adjustment disorder (AD) is a common and under-researched condition. In particular there is a need to consider in a transplant population where there is a significant stressors. This study is novel in considering a relative of the carer in parallel.

Main point:

This study finds that ~10% of the transplant population meet criteria on ADMN for AD, but this is a screening measure as opposed to a diagnostic measure (using either semi-structured interview or clinical diagnosis). Given that the mean time since transplant is 13.5 years when the cut-off for AD is 2years, this is unlikely to be AD, but instead some other phenomenon such as depressive episode or an anxiety disorder. The biggest limitation of this study is lack of any convincing diagnosis, and absence of any measure of depressive or anxiety symptoms, particularly when depression has been found to be present in post-transplant patients with a similar prevalence rate of up to 63% [1-4]. Given that one of the biggest controversies in AD research is the difficulty in distinguishing it from depression, this is certainly a point that is worthy of mention [5]. I remain unconvinced that AD was measured in this study.

Response: We agree with the reviewer with regard to the limitations of self-report questionnaires and added the lack of a semi-structured interview or clinical diagnosis to the limitations section of the manuscript. With regard to the validity of the symptoms assessed by the ADNM, however, we are confident that they are specific for the clinical picture of AD. The ADNM questionnaire is a well-validated measure that captures unique symptoms that represent ICD-11 AD. Its discriminant validity regarding depression and anxiety disorders (Bachem, Perkonigg, Stein, & Maercker, 2016; Einsle, Köllner, Dannemann, & Maercker, 2010; Maercker, Einsle, & Kollner, 2007) as well as general psychological distress (Lorenz, Hyland, Maercker, & Ben-Ezra, 2018) had been evaluated in several previous studies. Since the items of the questionnaire were presented in a contextualized form, referring explicitly to the patient’s organ transplantation, we believe that they capture adjustment symptoms specifically related to the transplantation. Nevertheless, according to ICD-11, depression (but not anxiety disorder) is an exclusion diagnosis for AD. Therefore, we now also present the prevalence rates for AD after excluding those cases. The prevalence dropped from 10.7 to 8.6% among patients and from 16.4 to 12.9% among patients. Other exclusion disorders of the ICD-11 are prolonged grief disorder, uncomplicated bereavement, burn-out, and acute stress reaction. We now mention the lack of a measure for these conditions in the limitations section.

References

Bachem, R., Perkonigg, A., Stein, D. J., & Maercker, A. (2016). Measuring the ICD-11 adjustment disorder concept: Validity and sensitivity to change of the Adjustment Disorder - New Module questionnaire in a clinical intervention study. International Journal of Methods in Psychiatric Research, 26(4), 1–9. doi:10.1002/mpr.1545

Einsle, F., Köllner, V., Dannemann, S., & Maercker, A. (2010). Development and validation of a self-report for the assessment of adjustment disorders. Psychology, Health & Medicine, 15(5), 584–595. doi:10.1080/13548506.2010.487107

Lorenz, L., Hyland, P., Maercker, A., & Ben-Ezra, M. (2018). An empirical assessment of adjustment disorder as proposed for ICD-11 in a general population sample of Israel. Journal of Anxiety Disorders, 54, 65–70. doi:10.1016/j.janxdis.2018.01.007

Maercker, A., Einsle, F., & Kollner, V. (2007). Adjustment disorders as stress response syndromes: A new diagnostic concept and its exploration in a medical sample. Psychopathology, 40(3), 135–146. doi:10.1159/000099290

General comments:

Both the title and the abstract were, on the whole, well written and reflect the content of the paper proper. The introduction gives a good overview of the literature and where this study fits into this.

Response: Thank you for the positive comments.

The methods give a detailed overview of the study. Cross-sectional study Participants were recruited using self-reported questionnaires - a non-clinical population. The measures and the statistical analysis were appropriate and well described, although there were some critical gaps: the lack of diagnosis (clinical or semi-structured), and absence of any measure of depressive or anxiety symptoms

Response: As mentioned above, we now consider this limitation and have added prevalence rates when depression is handled as an exclusion diagnosis.

The sociodemographic and clinical characteristics were described in the 'methods' section. Would these not be more appropriately detailed under 'results'?

Response: As suggested, the sociodemographic and clinical characteristics are now presented in the beginning of the ‘results’.

The results give a brief description of the findings.

The discussion provides a reasonable summary of this study, and covers the area of comparison between this study and the existing literature well. The limitations are discussed in reasonable detail, and the authors consider the recruitment bias inherent to its methodology. However, the most obvious limitation remains unconsidered – the lack of diagnosis, and absence of any measure of depressive or anxiety symptoms.

Response: These points were added to the limitations section.

The conclusion is appropriate to the findings of the paper, although I do not believe that the finds of one cross-sectional study support the final sentence “Should these findings be replicated in a study of a longitudinal design, AD’s time criterion of two years should be reconsidered.” This sentence would be best omitted.

Response: We agree with the reviewer and softened this sentence. It now reads: “Finally, it is noteworthy that the symptom burden of AD may persist in the long-term course after organ transplantation. This result implies that AD’s time criterion of two years should be evaluated in further studies of a longitudinal design.”

The references are profuse, relevant and up to date.

On the whole the tables and figures, are clear. In Table 1, mean and range for time since transplantation would have been a welcome addition.  Table 4 is titled “Hierarchical Regression to Predict AD among Patients”, when presumably it should read: “Hierarchical Regression to Predict AD among Relatives”.

Response: Thank you for pointing this out. We amended the title of the Table.

Minor typographic errors throughout, especially multiple counts of “infect” for “infection”.

Response: This was corrected and the paper was proof-read by a professional English editor.

References

 Dew MA, DiMartini AF. Psychological disorders and distress after adult cardiothoracic transplantation. J Cardiovasc Nurs. 2005; 20(5 Suppl): S51.  Fusar-Poli P, Lazzaretti M, Ceruti M, et al. Depression after lung transplantation: causes and treatment. Lung. 2007; 185: 55. Zalai D, Szeifert L, Novak M. Psychological distress and depression in patients with chronic kidney disease. Semin Dial. 2012; 25: 428. Dew MA, Rosenberger EM, Myaskovsky L, et al. Depression and Anxiety as Risk Factors for Morbidity and Mortality After Organ Transplantation: A Systematic Review and Meta-Analysis. Transplantation. 2015; 100: 988–1003. Bachem R, Casey P. Adjustment disorder: A diagnosis whose time has come. J Affect Disord, 2018; 227: 243-253.

Reviewer 2 Report

This is an excellent paper on an important topic that is understudied.  

i do believe two additional citations may help with the background and discussion.  the topic of coping should be included when discussing adjustment disorders.  both transplant patients and their relatives have better adjustment when they are more capable of problem solving, optimistic attitude, and or use religious coping (Tix and Frazier, 1998).  additionally much has been described about the different types of adjustments that occur in acute illness and chronic illness.  The discussion could include (Olbrisch 2002) comments on the adjustment back to a "state of health" where the patient's transplant has been successful in conventional terms but they continue to struggle with emotional and behavioral problems consistent with an adjustment reaction.  perhaps this concept has not been supported by your study where the patients with adjustment disorder seemed to be more associated with somatic symptoms (more health related concerns)

one minor editing change could be brought to the end of the background.   Three objectives are listed but in the preceding background description four topics are raised (adjustment diagnosis, transplant stressors/mental health, relatives, and social support).  the objectives should parallel this discussion.  #2 "transplant recipients/their relatives AD symtpoms" could be #3 with #4 as social support.  

in the limitations section you could add a comment about the response rate and sample size and how that could also impacted the prevalence of adjustment disorder.  ultimately 10% prevalence may be consistent with what one finds in a general medical clinic.   

Author Response

Response to Reviewer 2 Comments

This is an excellent paper on an important topic that is understudied.  

i do believe two additional citations may help with the background and discussion.  the topic of coping should be included when discussing adjustment disorders.  both transplant patients and their relatives have better adjustment when they are more capable of problem solving, optimistic attitude, and or use religious coping (Tix and Frazier, 1998).  additionally much has been described about the different types of adjustments that occur in acute illness and chronic illness.  The discussion could include (Olbrisch 2002) comments on the adjustment back to a "state of health" where the patient's transplant has been successful in conventional terms but they continue to struggle with emotional and behavioral problems consistent with an adjustment reaction.  perhaps this concept has not been supported by your study where the patients with adjustment disorder seemed to be more associated with somatic symptoms (more health related concerns)

Response: Thank you for these interesting inputs and references. We agree that the topic of coping is an important one in the field of adjustment disorders, which are at the threshold between normal and pathological stress reactions. However, as in the current study we did not assess coping in general or religious coping in specific as Tix and Frazier (1998) did, a substantial discussion of the topic of (religious) coping seems beyond the scope of this study. The conclusions section is dedicated to the potential clinical implications of the findings and we enriched it with a brief discussion of the role of coping resources for potential interventions focused on AD symptoms.

We were not able to find the Olbrisch (2002) study but did find a review by Olbrisch, Benedict, Ashe, & Levenson (2002) (reference below). We are not sure if this is the paper that was referred to by the reviewer. Our data does not include explicit information on whether the transplantation was considered successful or not - but given the long survival rates, this probably is the case for many participants. Even in the case of a successful transplantation, however, the presence of somatic symptoms and health-related concerns is the norm rather than an exception. Therefore, the results of our study are not necessarily at odds with the comments by Olbrisch (2002), as mentioned by the reviewer.

Olbrisch, M. E., Benedict, S. M., Ashe, K., & Levenson, J. L. (2002). Psychological assessment and care of organ transplant patients. Journal of Consulting and Clinical Psychology, 70(3), 771–783. doi:10.1037/0022-006X.70.3.771

One minor editing change could be brought to the end of the background.  Three objectives are listed but in the preceding background description four topics are raised (adjustment diagnosis, transplant stressors/mental health, relatives, and social support).  the objectives should parallel this discussion.  #2 "transplant recipients/their relatives AD symtpoms" could be #3 with #4 as social support.  

Response: The reason why there were three aims for the four topics in the introduction is that all of the aims are pursued for patients as well as for relatives. Rather than listing them separately for patients and relatives, we chose to combine them. However, we realized that indeed the aims of the study did not represent our explorations of symptom severity. We added a respective aim. The aims now read as follows:

The objectives of this study were: 1) to describe the prevalence of self-reported AD in a diverse sample of adult transplant recipients and their relatives and to compare AD and PTSD prevalence rates; 2) to explore the severity of AD symptoms among patients and relatives; 3) to assess the association of demographic-, transplant-, and health-related issues (i.e. somatic health, health-related quality of life) and transplant recipients/their relatives AD symptoms; and 4) to assess the role of social support with regard to AD symptoms among transplant recipients and their relatives.

In the limitations section you could add a comment about the response rate and sample size and how that could also impacted the prevalence of adjustment disorder.  ultimately 10% prevalence may be consistent with what one finds in a general medical clinic.   

Response: These points are now mentioned among the limitations of the study.

Reviewer 3 Report

This paper presents an analysis of dyadic data regarding adjustment disorder among organ transplant recipients and their family caregivers. Strengths include its use of an underserved clinical population and collection of data from matched patient-caregiver dyads. Overall, this is a well-conducted and compelling study, but I do have a few comments aimed at improving the paper.

The AD scale is used both to classify participants as having or not having AD and also as a continuous scale indicating AD severity. Although this does make sense intuitively, it would be useful to support the validity of this approach, if possible. This is especially important in this case because, if I am understanding the AD criteria correctly, is not based on a single numeric threshold on the scale – rather, the scale score of an individual identified as having AD could be lower than some individuals identified as not having AD, depending on the specific domains within which they met the qualifying criteria. I think more explanation of the scale scores versus AD criteria would help to demonstrate that it is justified to use the measure in both of these ways.

The rationale for including PTSD in the study is not completely clear, since these variables are analyzed less fully than the others – at least the means and SDs should probably be reported.. I can see some value in including this for comparison purposes with previous research, and to establish that AD is more appropriate diagnosis for many people in this population, but more explanation would be helpful to support these aims more explicitly.

Regarding the difference between patient and caregiver groups in terms of specific AD subscales, it is also important to note that there is a substantial difference in the gender composition of the two groups (patients are more likely to be male and caregivers are more likely to be female) which may partially explain the difference between patients and caregivers as groups. This should be addressed as a limitation, or possibly with supplemental analyses controlling for gender.

Author Response

Response to Reviewer 3 Comments

This paper presents an analysis of dyadic data regarding adjustment disorder among organ transplant recipients and their family caregivers. Strengths include its use of an underserved clinical population and collection of data from matched patient-caregiver dyads. Overall, this is a well-conducted and compelling study, but I do have a few comments aimed at improving the paper.

The AD scale is used both to classify participants as having or not having AD and also as a continuous scale indicating AD severity. Although this does make sense intuitively, it would be useful to support the validity of this approach, if possible. This is especially important in this case because, if I am understanding the AD criteria correctly, is not based on a single numeric threshold on the scale – rather, the scale score of an individual identified as having AD could be lower than some individuals identified as not having AD, depending on the specific domains within which they met the qualifying criteria. I think more explanation of the scale scores versus AD criteria would help to demonstrate that it is justified to use the measure in both of these ways.

Response: Thank you for pointing out this important issue. We decided to use the algorithm presented in the paper in order to approximate an ICD-11 diagnosis of AD, which requires that both main symptom clusters (preoccupations and failure to adapt symptoms) as well as functional impairment are present. This algorithm was developed by the authors of the ADNM-20 and has been used in several previous studies to approximate an AD diagnosis (eg., Bachem & Maercker, 2016; Ben-Ezra et al., 2018; Glaesmer, Romppel, Brähler, Hinz, & Maercker, 2015). There is also a numeric cut-off value for the ADNM which can be used in order to estimate a probable diagnosis of AD (Lorenz, Bachem, & Maercker, 2016). This cut-off value was developed based on the diagnostic algorithm mentioned above. However, when using the numeric cut-off value, very high symptoms in only one cluster but low symptoms in the other cluster might result in a diagnosis of AD, or AD might be attributed when functional impairment is not present – which would be inconsistent with the ICD-11 description. In order to represent prevalence rates, the algorithm is therefore more suitable. Symptom severity, however, is assessed with the continuous score and indeed there may be some cases that suffer from high symptom severity but not fulfill the AD criteria. However, such a discrepancy is due to the categorical nature of the diagnostic systems we use (ICD-11, DSM-5). The ADNM has been previously been used as a dimensional tool for assessing AD severity, for example in studies validating its convergent and discriminant validity (Einsle, Köllner, Dannemann, & Maercker, 2010; Maercker, Einsle, & Köllner, 2007).

We now provide a clearer rational for using the diagnostic algorithm for establishing AD prevalence rather than a numerical cut-off in the measure description.

References

Bachem, R., & Maercker, A. (2016). Self-help interventions for adjustment disorder problems: A randomized waiting-list controlled study in a sample of burglary victims. Cognitive Behaviour Therapy, 45(5), 397–413. doi:10.1080/16506073.2016.1191083

Ben-Ezra, M., Karatzias, T., Hyland, P., Brewin, C. R., Cloitre, Marylène Bisson, J. I., Roberts, N. P., … Shevlin, M. (2018). Posttraumatic stress disorder (PTSD) and complex PTSD (CPTSD) as per ICD-11 proposals: A population study in Israel. Depression and Anxiety, 35(3), 264–274. doi:10.1002/da.22723

Einsle, F., Köllner, V., Dannemann, S., & Maercker, A. (2010). Development and validation of a self-report for the assessment of adjustment disorders. Psychology, Health & Medicine, 15(5), 584–595. doi:10.1080/13548506.2010.487107

Glaesmer, H., Romppel, M., Brähler, E., Hinz, A., & Maercker, A. (2015). Adjustment disorder as proposed for ICD-11: Dimensionality and symptom differentiation. Psychiatry Research, 229(3), 940–948. doi:10.1016/j.psychres.2015.07.010

Lorenz, L., Bachem, R., & Maercker, A. (2016). The Adjustment Disorder–New Module 20 as a screening instrument: Cluster analysis and cut-off values. The International Journal of Occupational and Environmental Medicine, 7(4), 215–22. doi:10.15171/ijoem.2016.775

Maercker, A., Einsle, F., & Kollner, V. (2007). Adjustment disorders as stress response syndromes: A new diagnostic concept and its exploration in a medical sample. Psychopathology, 40(3), 135–146. doi:10.1159/000099290

The rationale for including PTSD in the study is not completely clear, since these variables are analyzed less fully than the others – at least the means and SDs should probably be reported.. I can see some value in including this for comparison purposes with previous research, and to establish that AD is more appropriate diagnosis for many people in this population, but more explanation would be helpful to support these aims more explicitly.

Response: PTSD indeed was assessed mainly in order to compare AD and PTSD prevalence rates. By this, we hope to draw attention to the AD diagnosis in the context of organ transplantations. We now mention this more specifically in the first aim of the study: 1) to describe the prevalence of self-reported AD in a diverse sample of adult transplant recipients and their relatives and to compare AD and PTSD prevalence rates.

Furthermore, we now include PTSD symptoms in the correlation matrix of the study variables, where we also present means and standard deviations.

Regarding the difference between patient and caregiver groups in terms of specific AD subscales, it is also important to note that there is a substantial difference in the gender composition of the two groups (patients are more likely to be male and caregivers are more likely to be female) which may partially explain the difference between patients and caregivers as groups. This should be addressed as a limitation, or possibly with supplemental analyses controlling for gender.

Response: Thank you for this input. We conducted an additional analysis and indeed the difference for preoccupation symptoms between patient and caregiver was no longer significant. This is now mentioned in the results section. The conclusions section has been amended accordingly.